# Supercurrent diode effect and magnetochiral anisotropy in few-layer NbSe$_2$

Lorenz Bauriedl[1], Christian Bäuml[1], Lorenz Fuchs[1], Christian Baumgartner[1], Nicolas Paulik[1], Jonas M. Bauer[1], Kai-Qiang Lin[1], John M. Lupton[1], Takashi Taniguchi [2], Kenji Watanabe [2], Christoph Strunk [1] & Nicola Paradiso [1✉]

Nonreciprocal transport refers to charge transfer processes that are sensitive to the bias polarity. Until recently, nonreciprocal transport was studied only in dissipative systems, where the nonreciprocal quantity is the resistance. Recent experiments have, however, demonstrated nonreciprocal supercurrent leading to the observation of a supercurrent diode effect in Rashba superconductors. Here we report on a supercurrent diode effect in NbSe$_2$ constrictions obtained by patterning NbSe$_2$ flakes with both even and odd layer number. The observed rectification is a consequence of the valley-Zeeman spin-orbit interaction. We demonstrate a rectification efficiency as large as 60%, considerably larger than the efficiency of devices based on Rashba superconductors. In agreement with recent theory for super-conducting transition metal dichalcogenides, we show that the effect is driven by the out-of-plane component of the magnetic field. Remarkably, we find that the effect becomes field-asymmetric in the presence of an additional in-plane field component transverse to the current direction. Supercurrent diodes offer a further degree of freedom in designing superconducting quantum electronics with the high degree of integrability offered by van der Waals materials.

[1] Institut für Experimentelle und Angewandte Physik, University of Regensburg, Regensburg, Germany. [2] International Center for Materials Nanoarchitectonics, National Institute for Materials Science, Tsukuba, Japan. ✉email: nicola.paradiso@physik.uni-regensburg.de

The archetypal example of a nonreciprocal electronic device is the diode. The term nonreciprocity in this context is used to imply a large difference in resistance between opposite bias polarities. For a conventional semiconductor diode, nonreciprocity follows from the inequivalence of the two crystals forming the pn junction, which have different types of doping. In homogeneous devices, polarity-dependent resistance is observed when both inversion and time-reversal symmetry are broken simultaneously. As shown by Rikken et al.[1,2], in noncentrosymmetric conductors nonreciprocal resistance can be phenomenologically described by

$$R = R_0[1 + \alpha B^2 + \gamma BI], \tag{1}$$

where the coefficient $\alpha$ refers to the usual magnetoresistance and $\gamma$ is the magnetochiral anisotropy (MCA) coefficient. In normal conductors, $\gamma$ is typically very small. Its strength is determined by the ratio between spin–orbit perturbation and the Fermi energy. Nonreciprocal transport therefore becomes discernible in semiconductors with low Fermi level and large spin–orbit interaction (SOI)[3]. The MCA effect can be greatly amplified in non-centrosymmetric superconductors[4–6,6–8]. Here, the energy scale governing the fluctuation regime close to the critical temperature is not the Fermi energy, but the superconducting gap. Another way to boost MCA for the resistance is to engineer nonreciprocal vortex motion in a superconductor by asymmetric patterning of artificial pinning centers[9].

In the past years, superconductivity-enhanced MCA for the resistance has been studied extensively. There is a satisfactory understanding of the mechanisms producing nonreciprocal resistance and theory predictions successfully describe the experiments[10,11]. On this basis, some first applications, such as superconducting antenna rectifiers[8] or spin filtering diodes[12], have already been proposed.

So far, studies on nonreciprocal transport, even when exploiting superconductivity to enhance MCA, have mainly focused on resistance. The seminal demonstration of a dissipationless nonreciprocal supercurrent[13] was only recently provided by experiments on synthetic Rashba superconductors based on Nb/V/Ta multilayers[14] and on Josephson junctions[15] with strong Rashba SOI. Such a nonreciprocal supercurrent gives rise to the so-called superconducting diode effect, where the supercurrent can flow only in one direction that can be switched by a magnetic field.

The MCA for the electrical resistance and that for the supercurrent are two clearly distinct effects. The latter is characterized by the kinetic inductance, which is uneven in the current, while the resistance is zero. In the experiments reported so far, samples that showed nonreciprocal supercurrent far below $T_c$ also showed nonreciprocal resistance in the fluctuation regime near $T_c$[14,15]. This correlation between the two phenomena arises due to the fact that both effects require the same symmetry-breaking mechanisms (i.e., time and inversion symmetry). While there is a satisfactory understanding of MCA for the resistance, the theoretical study of magnetochiral effects in supercurrents is just in its infancy[16–25]. Nonreciprocal supercurrent is better understood in Josephson junctions, where the diode effect can be engineered by Andreev bound states in the normal weak-link. The observation of non-reciprocal supercurrent in Rashba superconductors raises the question about its existence in materials with other types of SOI. Promising candidates are transition metal dichalcogenides (TMDs) that feature valley-Zeeman SOI, where magnetochiral resistance was already measured, while non-reciprocal supercurrent was predicted[26] but not yet observed.

Here, we report on the observation of a pronounced supercurrent diode effect in constrictions of few-layer NbSe$_2$-crystals. Owing to dominant valley-Zeeman SOI, the supercurrent diode behavior is driven by the out-of-plane component $B_z$ of the magnetic field. Unexpectedly, also the in-plane component affects the non-reciprocal supercurrent: we find that it breaks the anti-symmetry of the critical current difference with respect to $B_z$, boosting the rectification for one $B_z$ direction, and suppressing it for the opposite one.

## Results

Our samples are fabricated using standard exfoliation methods for TMDs[27–30]. A scheme of the typical device is depicted in Fig. 1a. The exfoliated NbSe$_2$ crystals considered here are between 2 and 5 layers thick[29]. The flake thickness can be estimated with reasonable accuracy from the optical contrast of the crystal when stamped on standard SiO$_2$/Si substrates. The parity of the layer number $N$ and the lattice orientation is determined, a posteriori, by second harmonic generation (SHG) measurements for all samples. NbSe$_2$ crystals are fully encapsulated in hBN[30] and edge contacts are fabricated by electron beam lithography[31]. A 250 nm-long channel of 250 nm width is patterned by electron beam lithography and reactive ion etching. The etched parts appear as dark purple triangles in Fig. 1b. The purpose of the narrow channel is to have a well-defined direction of the current density in the constriction, whose direction is indicated by **I** in Fig. 1a. In what follows, the x-direction is assumed to be that of the constriction axis (and thus that of the supercurrent) while the z-direction is perpendicular to the sample plane.

The main measurements of this work (sample F, see below) were performed in a dilution refrigerator equipped with a 9 T-superconducting coil controlling the in-plane component of the magnetic field. Additional coils provide a field perpendicular to the sample plane. A piezo-rotator allows for rotation of the sample around an axis normal to the sample plane. Additional measurements (samples B–E and G, see Supplementary Information) were performed in a $^4$He cryostat with a base temperature of 1.3 K, equipped with a single superconducting coil. Figure 1c–e introduces the supercurrent diode effect as measured in a NbSe$_2$ constriction patterned on sample G. The graphs show three pairs of current–voltage characteristics (IVs) for applied out-of-plane magnetic fields of $B_z = 0$ mT (c), $B_z = 32.5$ mT (d) and $B_z = -32.5$ mT (e). Importantly, all the IVs reported here always refer to the zero-to-finite (either positive or negative) bias sweep direction, in order to rule out heating effects. In the three graphs, the black (red) symbols refer to current density in the narrow channel oriented towards the positive (negative) $\hat{x}$ direction. A strong supercurrent diode effect[14,15] is evident from the comparison of the IVs: there is a marked difference $\Delta I_c \equiv I_c^+ - |I_c^-|$ in the critical current for the two supercurrent orientations, the sign of which changes when the magnetic field $B_z$ is inverted. The complete field dependence of both $I_c^+$ (black) and $|I_c^-|$ (red) is plotted in Fig. 1f. The current range between $I_c^+$ and $|I_c^-|$ corresponds to the supercurrent diode regime, where current flows without dissipation only in one direction, which can be selected by changing the sign of the magnetic field[14]. As a figure of merit of the supercurrent diode effect, one can take the supercurrent rectification efficiency $Q \equiv 2\Delta I_c/(I_c^+ + |I_c^-|)$[18], i.e. the difference between $I_c^+$ and $I_c^-$ normalized by their average, which is plotted in Fig. 1g versus $B_z$. For moderate fields, $Q$ increases almost linearly with the field. The maximum magnitude of $Q$ is above 60%, much larger than that ($\approx 5\%$) observed in ref. [14]. Beyond a certain breakdown field $B_{\text{max},Q} \approx 35$ mT the diode effect is gradually suppressed. This behavior is reminiscent of that observed in Rashba superconductors, see, e.g. Fig. 2 in ref. [14] and Fig. 3 in ref. [15]. Theory models for Rashba superconductors[16,19] predict a similar suppression, but the threshold field is expected to be of the order of the paramagnetic limit, i.e. much larger than the value observed in our and in other

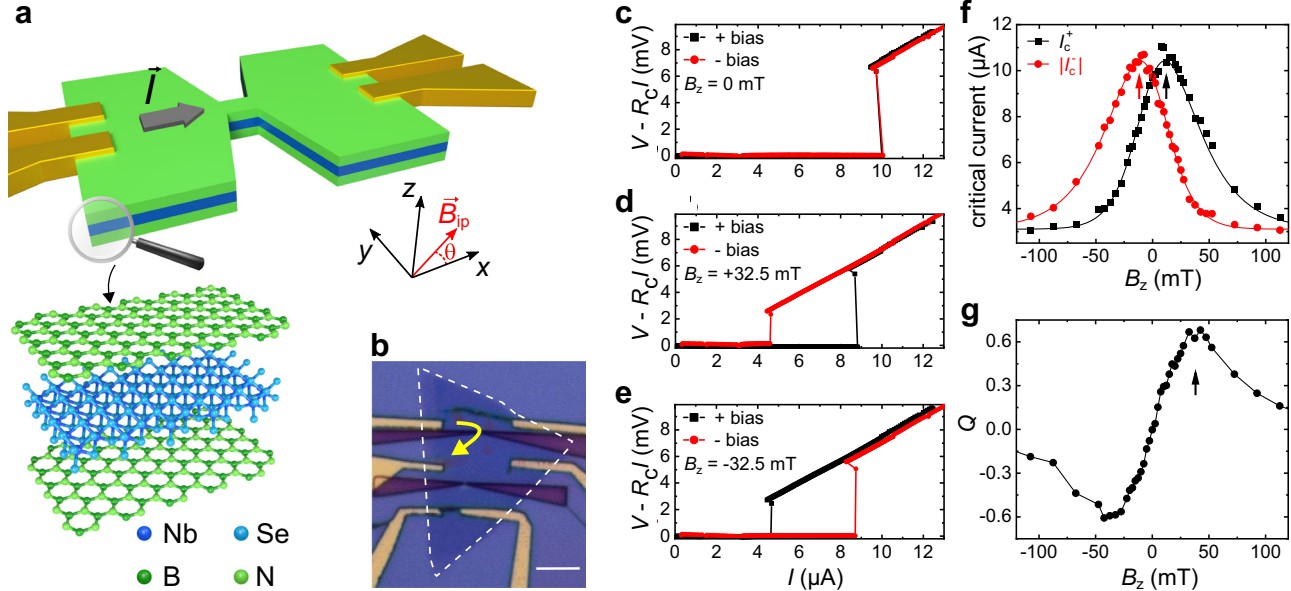

**Fig. 1 Supercurrent diode effect in a van der Waals superconductor. a** Scheme of the typical device. The central constriction is 250 nm wide and 250 nm long. The $x$-direction is chosen to be that of the supercurrent, i.e., the constriction axis. The $z$-direction is perpendicular to the crystal plane. The device is fabricated starting from a stack of hBN ($\approx$10 nm, tens of layers), NbSe$_2$ (2, 3 or 5 layers) and again hBN (tens of layers). **b** Optical micrograph of sample B. The dashed white contour highlights the NbSe$_2$ crystal. Electrodes are fabricated by edge contact techniques[27,31,47], while the constrictions are made by reactive ion etching. The yellow arrow indicates the supercurrent pathway. The scale bar corresponds to 5 μm. **c** Current–voltage characteristics (IVs) for opposite bias polarities (i.e., opposite current directions) measured on sample G in a 3-terminal configuration for zero out-of-plane field $B_z$. The sweep direction is always from zero to finite bias. A contact resistance $R_c = 1\,\mathrm{k\Omega}$ has been subtracted. **d** Similar measurements, but for $B_z = 32.5$ mT. Notice the difference between the two critical currents. **e** Same as in panel (**d**), but with opposite field orientation. Notice that the role of the two bias polarities now is swapped. **f** Absolute critical current for positive (black) and negative (red) bias as a function of $B_z$. Each value is the average of 10 consecutive measurements. The critical current is maximal for a nonzero $B_z$, namely for $|B_z| = B_{\mathrm{max},I_c} \approx 10$ mT (black and red arrow). The red and black solid line are guides to the eye, mutually symmetric upon reflection around $B_z = 0$. **g** Supercurrent rectification efficiency $Q \equiv 2(I_c^+ - |I_c^-|)/(I_c^+ + |I_c^-|)$, plotted versus $B_z$. $Q$ is maximal for $B_z = B_{\mathrm{max},Q} \approx 35$ mT (arrow). Measurements in (**c**–**g**) were performed at 1.3 K. As discussed in the Supplementary Information, an offset of $-2.5$ mT and 0.17 μA has been removed from $B_z$ and $I_c$, respectively.

experiments[14,15]. Finally, we stress that the observations in Fig. 1f, g do not depend on the field sweep direction, as discussed in the Supplementary Information.

We notice that, for either of the two bias polarities, the critical current increases by increasing the magnetic field, reaching a maximum at a nonzero field $B_{\mathrm{max},I_c}$. This can be seen, e.g., in Fig. 1f, where $B_{\mathrm{max},I_c} \approx 10$ mT. The remarkable increase of the critical current (the origin of which certainly deserves further study) is important since it eliminates the possibility that the nonreciprocal supercurrent originates from Joule heating due to, e.g., nonreciprocal resistance, which in NbSe$_2$ is known to be important near $T_c$ owing to magnetochiral effects[8]. A very recent work[32] on V or Nb superconducting films reported a bias polarity-dependent critical current increase, which gives rise to a $I_c^\pm(B)$ dependence similar to that in Fig. 1f (i.e., an inverted-W-shaped graph). The authors of ref. [32] attributed the phenomenon entirely to the interplay between Meissner currents and barrier for vortex entry[33]. In our work, however, the diode effect is not necessarily bound to the critical current increase: for instance, in sample G and D the rectification efficiency $Q$ keeps increasing well beyond $B_{\mathrm{max},I_c}$ (up to $B_{\mathrm{max},Q}$, which is more than three times $B_{\mathrm{max},I_c}$, cf. Fig. 1f, g), while in sample F, discussed below, the critical current increase is simply absent ($B_{\mathrm{max},I_c} = 0$). As discussed in the Supplementary Information, the experimental evidence seems to exclude a contribution of vortices or screening currents (the constriction width is comparable to the penetration depth $\lambda$ and much smaller than the Pearl length $\lambda^2/d$). However, the remarkable results reported in ref. [32] urge the use of caution

in interpreting the origin of supercurrent nonreciprocity in the presence of perpendicular fields.

We observed supercurrent rectification in other devices with the same nominal geometry. The supercurrent diode effect was clearly visible in all the samples where supercurrent could be measured, i.e., samples B, D, E, F, and G. Their layer number $N$ was determined by combining information from white-light optical and SHG microscopy[34–37], as discussed in the Supplementary Information. We found a layer number $N = 3$ for samples B, D and F, $N = 5$ for sample E and $N = 2$ for sample G. As for the Ising superconductivity, the supercurrent diode effect is not theoretically expected to occur in NbSe$_2$ when $N$ is even, owing to the restored inversion symmetry. However, both Ising superconductivity (see Fig. 4 in ref. [34]) and the supercurrent diode effect (this work) are experimentally observed for both even and odd $N$. This fact is likely caused by the relatively weak electronic coupling between the layers, which effectively renders NbSe$_2$ a collection of monolayers[34].

The experimental results shown in Fig. 1 clearly demonstrate that an out-of-plane magnetic field is required for the supercurrent diode effect in materials with valley-Zeeman SOI. However, it is interesting to study whether the in-plane field also affects the rectification. For this reason, we measured one device, sample F, in a setup equipped with a piezo-rotator. This setup allows us to rotate the sample with respect to the main magnetic field $\mathbf{B}_{ip}$, and therefore to control the angle $\theta$ between $\mathbf{B}_{ip}$ and supercurrent $\mathbf{I}$, as indicated in Fig. 1a. Additional coils, perpendicular to the main one, provide an independent control of the

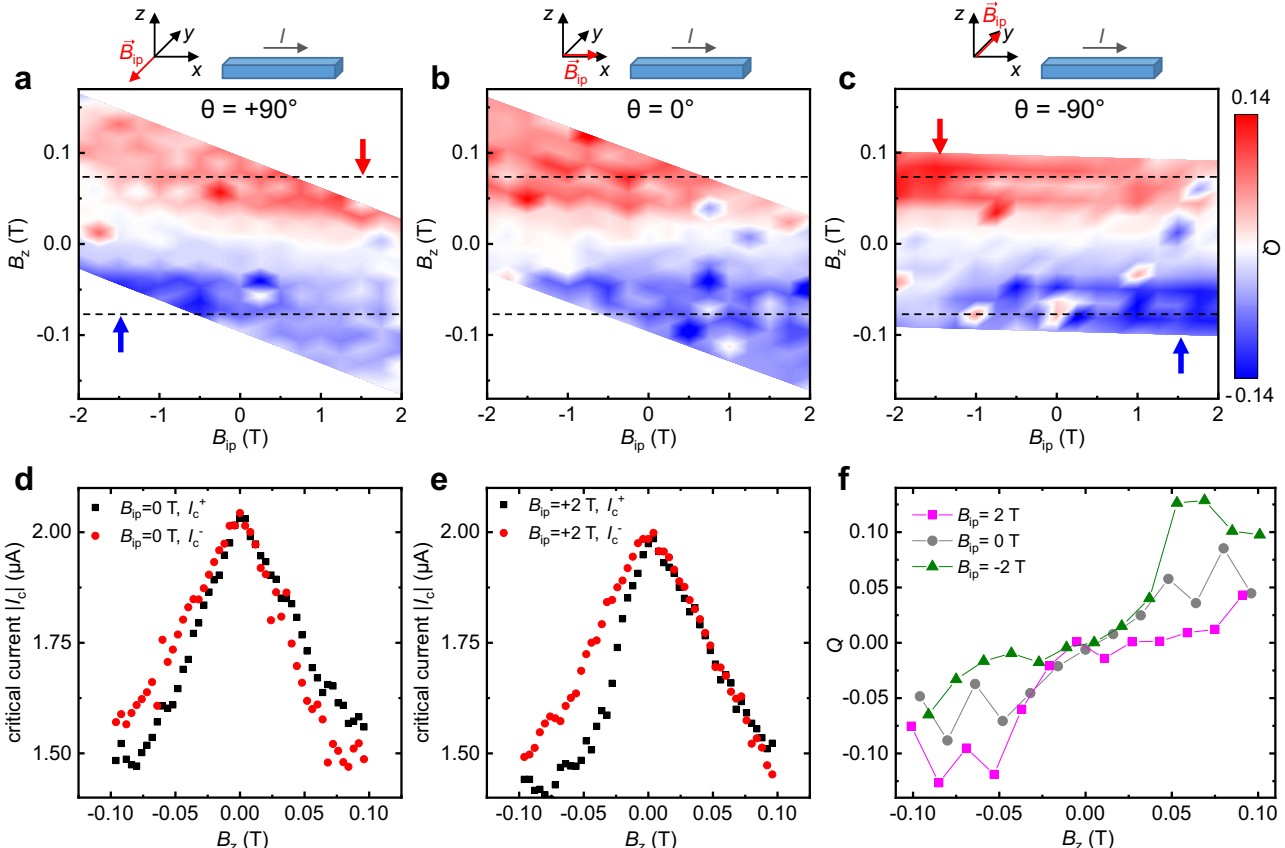

**Fig. 2 Supercurrent diode effect versus in- and out-of-plane field. a** The color plot shows $Q \equiv 2(I_c^+ - |I_c^-|)/(I_c^+ + |I_c^-|)$ as a function of out-of-plane ($B_z$) and in-plane ($B_{ip}$) field, measured in sample F for $\theta = 90°$ (i.e., for $\mathbf{B}_{ip} \perp \mathbf{I}$). $B_z$ here includes both the field produced by the orthogonal coils and the finite $z$-component of $\mathbf{B}_{ip}$ arising due to misalignment. Red and blue arrows indicate the areas where the diode effect is enhanced. **b** Similar measurements, but for $\theta = 0°$. **c** As in (**b**), but for $\theta = -90°$. Notice that this graph can be mapped onto that in (**a**), provided that $B_{ip} \to -B_{ip}$. The color contrast is the same in (**a**), (**b**) and (**c**). **d** Absolute value of $I_c^+$ (black) and $I_c^-$ (red) plotted versus $B_z$, for $B_{ip} = 0$ and for the sample orientation $\theta = -90°$. **e** As in (**d**), but for $B_{ip} = 2$ T. Note that data in (**d**, **e**) were measured in a different session (with higher resolution in $B_z$) compared to data in **c**. **f** Supercurrent rectification efficiency $Q$ as a function of $B_z$ at $\theta = -90°$ for $B_{ip} = -2$ T (green), 0 T (gray), and 2 T (magenta). Here, we used the same data as in panel **c**, for $B_{ip}$ values indicated in the legend. For $B_{ip} = 0$ we have substituted three outliers (for $B_z = -48$, $-64$ and $-80$ mT) with the corresponding values for the adjacent in-plane field $B_{ip} = -0.25$ T. For $B_{ip} = -2$ T we have substituted one outlier (for $B_z = -43$ mT) with the corresponding value for the adjacent in-plane field $B_{ip} = -1.75$ T, see Supplementary Information. Outliers are also visible in panels (**a**–**c**). All measurements reported in this figure were performed at 500 mK.

out-of-plane field $B_z$. Figure 2a–c show the supercurrent rectification efficiency $Q$ in sample F, plotted as a function of both the in-plane ($B_{ip}$) and the out-of-plane ($B_z$) magnetic field, for $\theta = 90°$ (panel (**a**), $\mathbf{B}_{ip}$ transverse to the supercurrent), for $\theta = 0°$ (panel (**b**), $\mathbf{B}_{ip}$ parallel to the supercurrent), and for $\theta = -90°$ (panel (**c**), $\mathbf{B}_{ip}$ antiparallel to that in panel (**a**). Owing to sample misalignment, a large in-plane field produces a significant out-of-plane component. This misalignment can be quantified (and accounted for, as in Fig. 2a–c) by looking at the maximum of $I_c^+(B_z)$ [or $I_c^-(B_z)$], as discussed in the Supplementary Material. As a result of this offset adjustment, the accessible range in the ($B_{ip}$, $B_z$) plane appears as a rhomboid.

The most prominent feature in these measurements is the vertical gradient in the color plot, indicating that the rectification efficiency increases monotonically with $B_z$ and is zero when $B_z = 0$, at least within the experimental data scatter. Hence, we have now established experimentally that, as predicted by theory[26], the supercurrent diode effect in TMDs requires an out-of-plane field. In contrast, it is the in-plane field that drives the diode effect in Rashba superconductors. Moreover, unlike what is predicted for superconductors with pure valley-Zeeman SOI[26], we observe that the in-plane field does affect the supercurrent diode effect. More precisely, the in-plane field component

perpendicular to the current breaks the (anti)symmetry of $Q$ as a function of $B_z$. If $B_{ip,y} \equiv \mathbf{B}_{ip} \cdot \hat{\mathbf{y}} = 0$ then $Q(B_z) = -Q(-B_z)$, with no dependence on $B_{ip,x} \equiv \mathbf{B}_{ip} \cdot \hat{\mathbf{x}}$, see Fig. 2b. Instead, Fig. 2a, c prove that, for finite $B_{ip,y}$, $Q(B_z) \neq -Q(-B_z)$, i.e., the diode effect becomes asymmetric in $B_z$. This effect is apparent, e.g., in Fig. 2a: for the given field and current orientation, the diode effect is enhanced when $B_z$ and $B_{ip}$ are both positive or negative (first and third quadrant of the graph, deep red and deep blue region, indicated by the arrows). On the other hand, it is suppressed when $B_z$ and $B_{ip}$ have opposite signs (second and fourth quadrant, light red and light blue region). At fixed $B_z$ (dashed lines), inverting the sign of $B_{ip,y}$ corresponds to a transition from enhanced to suppressed diode effect, i.e. from deep blue to light blue for the upper dashed line. Clearly, a 180° sample rotation is equivalent to a sign change in $B_{ip}$, as it is evident by comparing Fig. 2a, c. It is important to remark that data in Fig. 2a–c are still (anti)symmetric upon simultaneous inversion of both $B_z$ and $B_{ip,y}$, i.e., $Q(B_z, B_{ip,y}) = -Q(-B_z, -B_{ip,y})$.

To better visualize the impact of the in-plane field, it is instructive to look separately at $I_c^+$ and $I_c^-$ as a function of $B_z$, both with and without in-plane field. The latter case is shown in Fig. 2d. Both $I_c^+(B_z)$ and $I_c^-(B_z)$ appear as asymmetric, Λ-shaped functions. Their difference in slope for positive and negative $B_z$

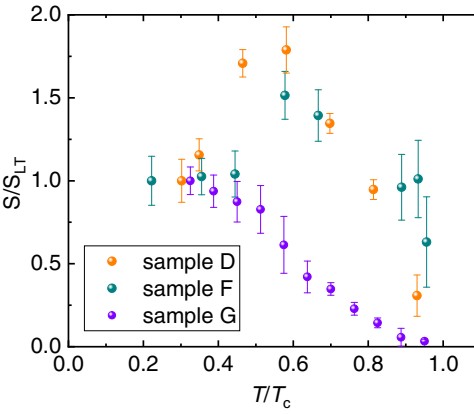

**Fig. 3 Temperature dependence of the supercurrent diode effect.** The graph shows the temperature dependence of $S \equiv Q/B_z$ for sample D, F, and G. $S$ is evaluated from linear fits of $Q(B_z)$ near $B_z = 0$, i.e. in the linear regime of the supercurrent rectification efficiency, where $Q \propto B_z$. The error bars correspond to the standard error of the fits. The $S$ values are normalized to the lowest temperature value $S_{LT}$ (e.g. 17.2 T$^{-1}$ for sample G), while the temperatures are normalized to the critical temperature $T_c$ of the corresponding sample. Orange, blue and purple symbols refer to sample D, F and G, where the critical temperature of the constriction region is 4.3 K, 2.25 K and 4.0 K, respectively.

produces a supercurrent diode effect increasing monotonically with $B_z$. We note the similarity of Fig. 2d and Fig. 1f with the corresponding measurements on Rashba Josephson junctions, see Fig. 3e in ref. [15]. As for Fig. 1g, the whole graph in Fig. 2d is symmetric if both the sign of $B_z$ and the direction of the supercurrent are changed simultaneously, i.e. for $B_z \leftrightarrow -B_z$ and $I_c^+ \leftrightarrow I_c^-$. This symmetry is broken when an in-plane field is applied along the direction perpendicular to the supercurrent. In the case of $B_{ip,y} > 0$, shown in Fig. 2e, the difference between the slope of $I_c^+(B_z)$ and $I_c^-(B_z)$ is reduced for $B_z > 0$ and enhanced for $B_z < 0$, and vice versa for $B_{ip,y} < 0$.

Figure 2f summarizes our main findings: the supercurrent rectification efficiency $Q$ is plotted as a function of $B_z$ for $\theta = -90°$ and for $B_{ip} = 2$ T, 0 T and $-2$ T. The graph shows that the perturbation due to the in-plane field is evident only for sufficiently high out-of-plane field, i.e., $|B_z| > 30$ mT. Above that threshold, for $B_z > 0$, the slope of $Q$ versus $B_z$ increases strongly for $B_{ip} = -2$ T and decreases for $B_{ip} = 2$ T. The opposite is true for $B_z < 0$. We also note that at sufficiently large $|B_z|$, above approximately 50 mT, the diode effect starts to be suppressed, so that at around $|B_z| \gtrsim 100$ mT the different curves in Fig. 2f merge again.

The key result of our observations is that the supercurrent diode effect in NbSe$_2$ is controlled by the out-of-plane field, as predicted by theory for superconducting TMDs[26]. For $B_z = 0$ there is no diode effect, independent of $\mathbf{B}_{ip}$. On the other hand, if $\mathbf{B}_{ip}$ has a component $B_{ip,y}$ perpendicular to the current, then the supercurrent diode effect becomes asymmetric in $B_z$, i.e., it is enhanced for one out-of-plane field polarity and suppressed for the other. The role of the two polarities is swapped if both the sign of the supercurrent and the sign of $B_{ip,y}$ are inverted. To the best of our knowledge, there are no predictions to date regarding a possible role of the in-plane field on the supercurrent diode effect in TMDs. In contrast, for Rashba superconductors it is precisely the in-plane field that controls MCA and the diode effect[14,15]. It is clear that further studies are needed to elucidate the role of the in-plane field in superconducting TMDs, which might be related to a possible Rashba-like component of the SOI[38,39]. In NbSe$_2$ such Rashba components can arise, e.g., due to

ripples in the crystal[40] formed when stamping NbSe$_2$ on hBN[30], or due to the substrate[41]. The existence of a weak Rashba SOI component in NbSe$_2$ has, for example, been invoked to explain the apparent two-fold anisotropy of the magnetoresistance as a function of in-plane magnetic field[41]. The model proposed in ref. [41] relies on the presence of $p$-wave components in the pairing function, which might as well play a role in the anisotropy of the diode effect we observe here.

Next, we turn to the temperature dependence of $Q$. Theory predictions regarding the temperature dependence of the rectification efficiency are quite diverse. While some models predict a square-root-like dependence near the critical temperature[18], others suggest a more complex functionality[19]. Figure 3 shows the temperature dependence of the diode effect in samples D, F and G. In the graph we plot $S \equiv Q/B_z$, obtained from $Q$ in the linear regime of low field, where $Q \propto B_z$. We choose to display $S$ rather than $Q$ because it represents a linear interpolation of $Q(B_z)$ and thus it averages over several data points, resulting in a reduced scatter compared to $Q$ for a given $B_z$.

First of all, we observe that the supercurrent diode effect saturates at low temperature. This saturation is compatible with all theory models and similar to the results on Josephson junctions in Rashba superconductors[15]. This result is not obvious, since in ref. [14] the diode effect is visible only near $T_c$ and it is suppressed for both higher and lower temperatures. We note also that the temperature dependence of the diode effect in samples D and F is nonmonotonic, with a maximum reached near $T/T_c = 0.5$. In contrast, a monotonic variation is observed in sample G. A nonmonotonic temperature dependence of the rectification at a finite field was predicted by recent theory[19]. It is possible that the different behavior displayed by samples D, F, and G is due to the fact that, in order to obtain a significant rectification, we need to apply a sufficiently strong field. Note that theory in ref. [19] predicts that the temperature dependence of the rectification is critically affected by a finite magnetic field.

## Discussion

A comment is in order about features which appear to be sample dependent. While the supercurrent diode effect was observed in every sample we measured (see Supplementary Information for further details), the maximum rectification efficiency strongly varies among the samples, ranging from 6 to 60%. Moreover, the supercurrent increase with $B_z$ is sample dependent, both in terms of relative increase $I_c(B_{\max,I_c})/I_c(0)$ and maximum supercurrent field $B_{\max,I_c}$. The supercurrent enhancement seems to be independent from the efficiency of the supercurrent rectification. The supercurrent increase is negligible in sample F, while in sample G no particular feature is observed in $Q(B_z)$ at $B_z = B_{\max,I_c}$. The nominal differences between the samples are the layer number and the orientation of $\mathbf{I}$ with respect to the lattice. Both features can be determined by SHG microscopy performed after the transport measurements, as described in the Supplementary Information. We found that neither the layer number nor the supercurrent-to-lattice orientation are correlated with the magnitude of the rectification efficiency. On the other hand, lithography of narrow constrictions by reactive ion etching produces random disorder, in particular at the edges. This randomness, together with the still limited number of studied samples, does not allow us to conclusively assess the role of the lattice orientation. Further investigation is needed in order to elucidate how the supercurrent diode effect is influenced by the direction of current flow with respect to the crystal.

In conclusion, we have demonstrated a supercurrent diode effect in few-layer NbSe$_2$. We show that the effect is controlled by the out-of-plane magnetic field, in contrast to what has been

observed in Rashba superconductors, where the effect is driven by the in-plane field component directed perpendicular to the supercurrent. This field component nevertheless plays a role in NbSe$_2$ devices as well, since it suppresses the diode effect for one out-of-plane field polarity and enhances it for the opposite one. Finally, the temperature dependence of the effect shows saturation at low temperature, a maximum or a kink at around $T = T_c/2$ and a suppression near $T_c$. TMD-based diodes may become crucial components in fully superconducting electronics. Their dissipation-free directional transport makes them suited for logic elements, ultrasensitive detectors, or signal demodulators, which can operate at low temperature with no energy loss. Being just a few atoms thick, their performance can conceivably be controlled electrically by gates, and modified by integrating them into complex van der Waals stacks in combination with other 2D materials.

During the review process we became aware of related experimental work on supercurrent rectification in NbSe$_2$/CrPS$_4$ heterostructures[42], in NbSe$_2$-based Josephson junctions[43], and on supercurrent rectification in magic-angle twisted bi-[44] and trilayer[45] graphene, and Nb-proximitized NiTe$_2$[46].

## Methods

**Sample preparation**. NbSe$_2$ crystals were purchased from HQ Graphene. hBN and NbSe$_2$ crystals were exfoliated following two common techniques. One is the technique introduced in ref. [28], where flakes are exfoliated on a poly-dimethylsiloxane (PDMS) film placed on a glass slide. Suitable crystals are then stamped onto the sample using a micromanipulator placed under a zoom lens. The other technique is commonly used for the production of fully hBN-encapsulated graphene devices[27]. In this case, flakes are sequentially picked up by a thick PDMS film coated with polycarbonate (PC). The flake pick-up takes place at 120°C, while the final release is triggered by melting the PC at 180°C and by dissolving it in chloroform.

Both the fabrication of the contacts and the design of the constriction require an electron beam lithography step followed by reactive ion etching (RIE). For the RIE, a mixture of 6 sccm O$_2$ and 40 sccm CHF$_3$ is ignited into a plasma with 35 W r.f. forward power at a pressure of 47 mbar. The RIE step etches through hBN and NbSe$_2$ with an approximate etching rate of 0.45 nm/s. When electrodes have to be produced, immediately after the RIE step a 10 nm-thick layer of Ti and a 100-nm-thick film of Au are deposited. This procedure reflects the well-known recipe for edge contact fabrication in graphene. For sample G the RIE process was substituted by an Argon plasma etching process with 2 kV acceleration voltage and 20 mA plasma current at about $3 \times 10^{-3}$ mbar. The etching rate is in this case approximately 1 nm per minute.

**Transport measurements**. Measurements on sample F were performed in a dilution refrigerator with a base temperature of 30 mK. For electrical measurements, we used DC lines with Cu-powder filters. Since one of the four electrodes stopped working after cool-down, we measured the device in a three-terminal configuration. The IV characteristics were thus obtained by subtracting the voltage $R_cI$, where $R_c = 413\ \Omega$ is the contact resistance. In the IV curves, the resistive transition of the constriction at the critical current was clearly visible as a sharp step, except very close to the critical temperature and field.

The critical currents for data in Figs. 1 and 3 were determined from the IVs as the extrapolation on the current axis (abscissas) of the steep voltage increase at the resistive transition. In Fig. 2, owing to the larger amount of data, we used the alternative criterion $V(I_c) = V_{\text{thres}} \equiv 100\ \mu V$, which is better suited for automatic routines. Nevertheless, the results shown here depend very weakly on the criterion for the critical current. We verified in all samples that the $I_c^{\pm}$ and $Q$ data obtained with either criterion were nearly the same (see Supplementary Information for further details).

Measurements on the other samples (A–E,G) were performed in a $^4$He cryostat with only room temperature filtering ($\pi$-filters). Except for sample G, the sample holder is positioned in such a way that the magnetic field produced by the superconducting coil is approximately in the plane of the sample surface. The misalignment (typically of the order of a few degrees) produces an out-of-plane field of tens of milliteslas per tesla of the applied field. In sample G, instead, the sample is perpendicular to the main field, thus the field is applied exclusively out-of-plane.

In our three- and four-terminal measurements we applied a voltage bias directly to the source and drain contacts, without using a preresistor (a resistance in series is provided by the Au-NbSe$_2$ contact resistance, which is of the order of 1 k$\Omega$). The voltage drop across the constriction and the current were then measured simultaneously.

**Second harmonic generation measurements**. Optical measurements of second harmonic generation (SHG) were always performed after transport measurements, in order to minimize the risk of photo-oxidation[30]. The co-polarized SHG intensity is measured as a function of the relative angle between laser polarization and crystal orientation[34–37]. The light source used was a pulsed Ti:sapphire laser (80 fs pulse duration, 80 MHz repetition rate, 1 mW power) at 800 nm. Using a microscope objective (×40, numerical aperture of 0.6), the light was focused onto the NbSe$_2$ samples placed in the vacuum chamber. The reflected SHG signal at 400 nm was collected with the same objective, filtered by a 680 nm short-pass filter, dispersed in a spectrometer (150 grooves/mm grating) and detected by a CCD camera. A linear polarizer was placed in front of the spectrometer to ensure acquisition of the signal polarized parallel to the laser polarization. A 50:50 non-polarizing beam splitter was used to separate the incident pathway from the signal detection pathway. In between the beam splitter and the objective, an achromatic half-wave plate was placed to change the relative angle between the crystal orientation and the laser polarization. The half-wave plate was rotated using a stepper motor. A 1200 grooves/mm grating was used as a reference sample. In the experiments, we used an exposure time of 1 s per data point.

**Reporting summary**. Further information on research design is available in the Nature Research Reporting Summary linked to this article.

## Data availability

The data that support the findings of this study are available at the online depository EPUB of the University of Regensburg, with the identifier https://doi.org/10.5283/epub.52406.

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

## Acknowledgements
The authors thank Denis Kochan, Marco Aprili, and Magdalena Marganska for useful discussions. The work was funded by the Deutsche Forschungsgemeinschaft (DFG, German Research Foundation)—Project-ID 314695032—SFB 1277 (subprojects B03, B04, B08, and B11), and by the European Union's Horizon 2020 research and innovation programme under grant agreements No 862660 QUANTUM E-LEAPS.

## Author contributions
N. Paradiso and C.S. conceived and designed the experiments. L.B., C. Bäuml, and N. Paulik fabricated the samples. L. B., L. F., C. Baumgartner and N. Paradiso, performed the transport experiments; L.B., J. M. L., K.-Q. L., and J. M. B. conceived and performed SHG measurements. K. W. and T. T. grew hBN crystals. All authors contributed to the preparation of the manuscript.

## Funding

## Competing interests
The authors declare no competing interests.
