## [Peer Review File · Nature Communications]

Supercurrent diode effect and magnetochiral anisotropy in few-layer NbSe₂REVIEWER COMMENTS

Reviewer #1 (Remarks to the Author):

This is an interesting paper which reports a supercurrent diode effect in NbSe₂ nanowires obtained by patterning NbSe₂ flakes. The authors demonstrate a rectification efficiency as large as 33%, considerably larger than the efficiency of devices based on Rashba superconductors reported so far. Since the study on the supercurrent diode effect has just started and the reports on it are still limited, this paper would stimulate the researchers in the field. However, understanding of the experimental result is not clear. I recommend revisions of the manuscript before the publication.

In Fig.1g, the authors point out the suppression of Q beyond a certain breakdown field ($B = 2$ T). They just mention that “such a suppression is remarkably similar to that observed in synthetic Rashba superconductors, e.g. Fig. 2 in Ref. [16] and Fig. 3 in Ref. [17]”.

Do the authors have any understanding or speculation of this suppression?

Although the authors claim in the abstract that they obtained a rectification efficiency as large as 33%, considerably larger than the efficiency of devices based on Rashba superconductors. However, the data for this large rectification efficiency is shown only in the Supplementary Material. It should be discussed in the main text if they insist the importance of the large Q.

The amplitude of the rectification efficiency Q is quite scattered from several % to 33 %. What determines the amplitude of Q?

Overall, the quality of the experimental results is not good. The data points are so scattered. I am wondering the reproducibility. Please show the results of multiple measurement for the same device.

Reviewer #2 (Remarks to the Author):

Currently superconducting (SC) diode effect is becoming a hot research topic. This work is a timely contribution on it. The authors report on realization of SC diode effect in few-layer NbSe₂ nanowires under an out-of-plane magnetic field. This is in contrast to the recently reported diode effect realized in the Nb/V/Ta superlattice and that in Josephson junctions, in which the SC diode effects were realized with in-plane magnetic fields. The authors also show that although the diode effect is mainly dominated by out-of-plane magnetic fields, an in-plane magnetic field can be used to tune the diode effect.

This manuscript is well written and the experimental results are very novel and original. In general, I would like to recommend publication of this paper in Nature Communications. However, there are several confusing points I would suggest the authors to address first.

1. Why is the odd layer number required for the diode effect in NbSe₂? I would appreciate a lot if the authors could describe the physical mechanism of the diode effect in some simple words in the manuscript even if it was described in the theoretical paper Ref. 21. Did the authors do any experiments using samples with even layer numbers? Did the diode effect disappear?

2. Since the diode behavior is driven by the out-of-plane magnetic field component, why is the experiment in Fig. 1 carried out in a magnetic field mostly directed in-plane? Would it be simpler and easier for understanding by directly show the effects and/or results under an out-of-plane magnetic field? In that case, the field values are meaningful.

Why the I_c curves in Fig. 1f have to be measured under two different temperatures? Would this induce the effect that the maximum I_c value is not at zero field?

What's the T_c of the samples? Usually, it's better to show RT curves (could be in the Supplementary Information) for a superconductor related experimental work.

3. The experiments of Fig. 2 measured on sample F were conducted by three-terminal configuration

because one of the four electrodes stopped working after cool-down. Would it be possible to repeat the results of Fig. 2 in a non-broken sample? This is because three terminal configuration experiments would include same effects from electrode contacts, which may not be simply corrected by subtracting a normal resistance component (the contact resistance may not be linear with current or temperature).

The method used to correct the field misalignment effect for the results in Fig. 2 is inconsistent with the main results in Fig. 1f. Figure 1f shows that the maximum I_c values are not at zero magnetic field. However, the authors corrected the misalignment effect for Fig. 2 by assuming that the maximum I_c values are at zero magnetic fields. This is really confusing.

4. The vortex motion was mentioned as one of the possible mechanisms for SC diode effect by Toshiya Ideue and Yoshihiro Iwasa in Nature News and Views [Nature 584, 249 (2020)] for the work of Ref. 16. Later, a SC diode effect based on vortex motion mechanism was reported in Ref. 9 with the sample in an out-of-plane magnetic field, the same with the dominant field orientation in this work. In general, one would expect emerging of vortices in a 2D superconductor in an out-of-plane magnetic field. Does the vortex motion play any or part of the role on the SC diode observed in this work? Although there are no symmetry breaking of vortex pinning sites (at least no artificial pinning sites) in the sample, however, the current flow with the two left current leads shown in Fig. 1b would not be uniformly straight along the sample channel (the channel is actually a square of 250 nm by 250 nm and is not long and 'narrow' considering the square geometry), as illustrated by the curved arrow line in Fig. 1b. This should lead to spatially symmetry breaking on the driving force to vortices. One possible experiment to exclude or confirm the possible vortex motion effect is to switch the current and the voltage leads. Applying current on the two right leads will reverse the symmetry breaking as compare to applying current on the two left leads. This will reverse the diode polarity for the vortex motion induced diode effect.

In fact, in a vortex motion related effect, the Bean-Livingston surface barrier [C. P. Bean and J. D. Livingston, Phys. Rev. Lett. 12, 14 (1964); E. Zeldov. et al., Phys. Rev. Lett. 73, 1428 (1994)] will lead to vortices entering and existing the sample not exactly at zero magnetic field. This effect is usually stronger for narrower samples. Would this effect related to the result of the maximum value of the I_c curves in Fig. 1f is not at zero magnetic field? Such effect can be easily excluded or confirmed by comparing results between two field sweeps (increasing and decreasing). For example, for a given current polarity, the maximum I_c value would be at positive (or negative) fields for increasing (or decreasing) fields, respectively. To be noted, reversing the field sweep direction would NOT reverse the diode polarity which is determined by the spatially symmetry-breaking.

The above comment does NOT mean the author's conclusion of valley-Zeeman SOI induced diode effect in NbSe₂ nanowires could be wrong. I recommend the authors to discuss and/or exam the possible role from vortex motion because the diode effect may not originate from a single mechanism.

In summary, the manuscript presents a very interesting and novel SC diode effect in few-layer NbSe₂ nanowires driven by out-of-plane magnetic fields. Given that the SC diode effect is significant for both future application and new physics, I will recommend publication of this work in Nature Communications if the authors could address my above comments/confusing properly.

REVIEWER COMMENTS

Reviewer #1 (Remarks to the Author):

This is an interesting paper which reports a supercurrent diode effect in NbSe2 nanowires obtained by patterning NbSe2 flakes. The authors demonstrate a rectification efficiency as large as 33%, considerably larger than the efficiency of devices based on Rashba superconductors reported so far.

Since the study on the supercurrent diode effect has just started and the reports on it are still limited, this paper would stimulate the researchers in the field. However, understanding of the experimental result is not clear. I recommend revisions of the manuscript before the publication.

We thank the Referee for his/her positive feedback and for his/her constructive criticism. We have now addressed in detail all the questions. Among many improvements, we would like to highlight that we have fabricated and measured a new sample, referred to as sample G, to further dissipate the doubts about the sample quality and reproducibility. This sample shows lower data scatter and a record rectification efficiency. These results confirm that 100% of the samples where Q was measured display the diode effect, validating the robustness of the effect and its high reproducibility. We will now address the questions of the Referee in detail below.

In Fig.1g, the authors point out the suppression of Q beyond a certain breakdown field ($B = 2$ T). They just mention that “such a suppression is remarkably similar to that observed in synthetic Rashba superconductors, e.g. Fig. 2 in Ref. [16] and Fig. 3 in Ref. [17]”.

Do the authors have any understanding or speculation of this suppression?

Before answering the question, a caveat: as stated in the manuscript, a solid theoretical model of this effect is still lacking at present. Moreover, as many theoretical groups are now actively working on this subject, the theoretical understanding is in continuous evolution, as it will become clear below. Clearly, this makes it evident how timely our experimental work is.

The current understanding of the effect mostly comes from the few phenomenological (i.e., based on the Ginzburg-Landau approach) models which have been proposed so far, see references [Yuan2021,Daido2022,He2021,Ilic2021] listed below. Importantly, these models focus on superconductors (SC) with Rashba spin-orbit interaction (SOI). We note that the only reference (He2021) also discussing SC with valley Zeeman SOI, removed that discussion in the last version of the preprint, which now focuses only on Rashba SOI.

That said, we shall try to answer the Referee’s question based on the existing models. All these models start from a generalized expression of the GL free energy which contains, instead of the simple coefficients α and β , polynomials in q , where q is the Cooper pair momentum. That is:

$$\alpha \rightarrow \alpha_0 + \alpha_1 q^1 + \alpha_2 q^2 + \dots \quad \text{and} \quad \beta \rightarrow \beta_0 + \beta_1 q^1 + \beta_2 q^2 + \dots \quad (1)$$

The non-zero Cooper pair momentum is an indirect consequence of the Zeeman perturbation on two Fermi surfaces with different spin orientation: the Zeeman term shifts the Fermi surfaces in k -space in such a way, that pairing occurs between states which do not have exactly opposite momentum. This implements a so-called helical superconducting state with Cooper pairs having a finite momentum q .

The nontrivial part of the analytical work of these references consists in deducing the coefficients from the Hamiltonian (containing the SC pairing, the SOI terms and the parts depending on the magnetic field) and then to calculate the critical currents (for each sign + or - of the supercurrent) from

$$j_{\pm} = 2 \frac{\partial F}{\partial q_{\pm}}.$$

In Rashba superconductors (SC diode effects have theoretically been studied almost exclusively on Rashba systems) there are two Fermi surfaces, one of them would favor a certain helical vector q , the other the opposite one, $-q$. The Fermi surface with the largest density of states dominates the effect and determines the sign of q for the helical state of the condensate. However, it turns out from theory calculations that diode effects emerge when both Fermi surfaces (with opposite q) contribute to SC pairing. This has been described for example in [Yuan2021,Ilic2021]. As a consequence, when the magnetic field becomes too large, only one Fermi surface dominates the pairing, and the diode effect is suppressed. To our knowledge, there is no intuitive and consistent picture from which this fact can be deduced: it simply results from independent calculations of different groups. Notice that paradoxically even an excessive increase of spin-orbit coupling would produce the same influence as a large field, suppressing the effect.

This is the result of (some) calculations on simplified systems. We believe that an empirical test of this fact might come from experiments on TI surface states, which implement the Rashba 2DEG with only one of the two Fermi surfaces. Owing to the preliminary nature of these results, and to the fact that they have so far been applied only to Rashba systems, we prefer not to discuss in detail these aspects in our experimental work. However, we have briefly mentioned them in the new version of the manuscript.

[Daido2021] A. Daido, Y. Ikeda, and Y. Yanase, Intrinsic superconducting diode effect (2021), arXiv:2106.03326. **Recently published as:** Phys. Rev. Lett. **128**, 037001 (2022).

[Yuan2021]: N. F. Q. Yuan and L. Fu, Supercurrent diode effect and finite momentum superconductivity (2021), arXiv:2106.01909. **Recently published as:** PNAS, **119**, e2119548119 (2022).

[He2021]: J. J. He, Y. Tanaka, and N. Nagaosa, A phenomenological theory of superconductor diodes in presence of magnetochiral anisotropy (2021), arXiv:2106.03575. **Recently published as:** *New J. of Phys.*, <https://doi.org/10.1088/1367-2630/ac6766> (2022).

[Ilic2021]: S. Ilic, F. S. Bergeret, Theory of the supercurrent diode effect in Rashba superconductors with arbitrary disorder (2022), arXiv:2108.00209.

Although the authors claim in the abstract that they obtained a rectification efficiency as large as 33%, considerably larger than the efficiency of devices based on Rashba superconductors. However, the data

for this large rectification efficiency is shown only in the Supplementary Material. It should be discussed in the main text if they insist the importance of the large Q .

The comment of the Referee is appropriate. Luckily, it turns out that the new sample we have measured (see comment above and below on sample G) has a record rectification efficiency, with $Q > 60\%$. Therefore, we now show data from this new sample in Fig.1. We would like to stress that this large rectification efficiency is quite remarkable and worth mentioning. However, it is not a central aspect of our work, which focuses instead on the physics of the SC diode effect in valley-Zeeman systems.

The amplitude of the rectification efficiency Q is quite scattered from several % to 33 %. What determines the amplitude of Q ?

This question is natural and important, and we have made a strong effort to provide an answer, which unfortunately remains elusive. Our initial guess was that the only thing that distinguishes the different samples (which are *nominally* identical from the geometry point of view) is the orientation of the supercurrent with respect to the NbSe₂ lattice. On the other hand, in an ideal, perfectly homogeneous, junction an out-of-plane field alone is not sufficient to break the symmetry between source- and drain-side of the constriction. Therefore, independently from the exact details of the supercurrent diode mechanism, one would expect that the (reciprocal) lattice orientation should play a role.

To answer this question, we conducted extensive work aimed at determining the crystallographic orientation of the sample with respect to the supercurrent direction. However, as discussed in the Supplementary Information, based on our SHG data it seems that the mutual orientation of lattice and supercurrent does not have a clear impact on the magnitude of the rectification efficiency Q . On the other hand, the number of samples we investigated is limited and they were fabricated independently. Therefore, accidental features of each individual constriction might mask the role of the lattice orientation. In fact, the only other characteristic that is sample dependent is the degree of disorder introduced by etching or by unseen defects in the crystal near or at the constriction - that is, by the unavoidable, random imperfections. However, disorder is hard to control, and it is difficult for us to quantify its contribution here.

Notably, we made significant efforts to reproduce the diode effect in multiple samples, despite the known difficulty in the preparation of such samples (TMDs are difficult to make electrical contact to, and, moreover, NbSe₂ tends to oxidize). It is precisely the repetition of the experiment on these samples that made it possible to identify features that are common (the very diode effect, the linearity of the rectification with B at small field, and its temperature dependence, as seen in Fig.3) and features that appear to be sample-dependent, such as the maximum rectification efficiency.

Overall, the quality of the experimental results is not good. The data points are so scattered. I am wondering the reproducibility. Please show the results of multiple measurement for the same device.

To answer in the most direct way this question, we provide below (Fig.R1) examples of multiple measurements on sample F. As an additional effort, we have fabricated and measured yet another

device (sample G, as mentioned above), with better results in terms of data scatter for Q . To better answer the question of the Referee, we have repeated the IV characteristics for the measurement of I_c^+ and I_c^- ten times for each B_z . The results are shown in Fig.R2 below (raw data). This data is the source for the new Fig.1f and Fig.1g. It is clearly visible that the switching current has a certain distribution, but this does not affect the conclusions of our work.

We would like to stress that the scatter of the data for the rectification Q is not synonymous of bad quality of the experiment, namely, noisy IV characteristics. For example, in the first version of our manuscript it was precisely the set of IVs with the largest noise level (sample B) that produced the least scattered $Q(B)$ curve, see previous Fig. 1g. Indeed, we empirically found that, as long as the noise (mostly voltage noise) is within a reasonable level of about 100-200 μ V, it does not affect the determination of I_c , which is robust, independently of the criterion for I_c (see Supp. Info section "Resistance and critical current versus temperature").

The data scatter comes from the fact that Q is obtained from ΔI_c which is a *difference* between two critical currents. A certain width in the distribution of the stochastic switching (the constriction can be modeled as a Josephson junction) is largely amplified when one plots differences. As a comparison, one can take the corresponding $Q(B)$ graphs in the seminal paper by Ando *et al.* Nature **584**, 373 (2020). A similar scatter of the data points is also observed there.

Moreover, as we discuss in the last question of Referee 2, it is also possible that vortices, while irrelevant for the determination of the diode effect (see answer to last question of Referee 2), do randomly affect either I_c^+ and I_c^- , producing large fluctuations in Q .

Now we shall display (as requested by the Referees) multiple measurements for the same device (we choose here sample F because of its central role in our work, and sample G where we purposely executed multiple measurements to elucidate this point).

Fig.R1: Three-terminal IV characteristics for sample F, measured at different values of B_z , indicated in the graph. Black (red) curves refer to positive (negative) bias, i.e., to supercurrent parallel (antiparallel) to the x axis as defined in the top-right scheme. Measurements for finite B_z have been placed in the second and third row in such a way that opposite values of B_z are vertically aligned. Since the scale of the graph is always the same, this allows the reader to capture at a glance the fact that inverting B_z inverts the role of positive and negative supercurrent, i.e. that $I_c^+(B_z) = I_c^-(-B_z)$. Notice that these two sets of measurements have been performed over several hours and with many other measurements in between. The matching $I_c^+(B_z) = I_c^-(-B_z)$ proves the reproducibility and stability of the measurements. The scatter in Fig.2 is mainly due to rare but strong outliers, whose origin we could not identify. Each IV is swept from zero bias to finite (positive or negative) bias. Data are taken from the same measurement set from which Fig.2 of the main text is taken. The low-bias slope refers to the contact resistance, which cannot be eliminated in a three-terminal measurement.

Fig. R2: Positive (upper side of the graph) and negative (lower) critical current measured as a function of the out-of-plane field B_z in sample G. For each value of B_z , several symbols (typically 10) are plotted which corresponds to repetitions of the IV measurement in the same conditions. A certain distribution of the critical current values is visible, which is responsible of a pronounced scatter in the plot for the difference between I_c^+ and $|I_c^-|$. These data are the basis for the plot in the new Fig. 1f of the main text, where for each B_z the average I_c is plotted. This graph appears in the new Fig.S1a.

Reviewer #2 (Remarks to the Author):

Currently superconducting (SC) diode effect is becoming a hot research topic. This work is a timely contribution on it. The authors report on realization of SC diode effect in few-layer NbSe2 nanowires under an out-of-plane magnetic field. This is in contrast to the recently reported diode effect realized in the Nb/V/Ta superlattice and that in Josephson junctions, in which the SC diode effects were realized with in-plane magnetic fields. The authors also show that although the diode effect is mainly dominated by out-of-plane magnetic fields, an in-plane magnetic field can be used to tune the diode effect.

This manuscript is well written and the experimental results are very novel and original. In general, I would like to recommend publication of this paper in Nature Communications. However, there are several confusing points I would suggest the authors to address first.

We thank Referee 2 for having pointed out the timeliness of our work and for his/her positive feedback. The constructive comments, together with those of Referee 1, motivated us to fabricate another sample (sample G, mentioned in the comments for Referee 1). Measurements on this sample were useful to clarify the doubts of the Referee and to corroborate the conclusions of the first version of the manuscript. We shall now discuss, point-by-point, all the questions of the Referee.

1. *Why is the odd layer number required for the diode effect in NbSe₂? I would appreciate a lot if the authors could describe the physical mechanism of the diode effect in some simple words in the manuscript even if it was described in the theoretical paper Ref. 21. Did the authors do any experiments using samples with even layer numbers? Did the diode effect disappear?*

In principle, and only *in principle*, the diode effect requires simultaneous breaking of time *and* inversion symmetry. The 2H-polytype of NbSe₂ (the one typically used in SC experiments) is non-centrosymmetric only for an odd number of layers, while it is centrosymmetric for an even number of layers. In *practice*, however, the layer number parity is less crucial than one would expect. The most striking example of this parity irrelevance is the Ising superconductivity effect [Xi2016]. While expected only for odd-layer-number-NbSe₂, it is observed for even layer numbers as well. The reason for this is that the coupling between adjacent layers (in terms of interlayer electron tunnel amplitude compared to spin-orbit coupling) is so weak that NbSe₂ behaves effectively as a collection of monolayers. (See e.g. Reference [Xi2016], Fig. 4 and discussion therein).

Coming to our work, at the beginning we have intentionally fabricated samples with odd-layer number, in order to maximize the chances to observe the effect. At the end, we had tried to fabricate one sample with even layer number. However, unfortunately, this last sample turned out to be odd-layer when analyzed *a posteriori* by SHG (it is not easy to know with certainty during the fabrication the layer parity from the optical contrast, since the typical error in this determination of the thickness is about one layer, which invalidates it for parity assessment).

The question of the Referee motivated us to try to **fabricate and measure a NbSe₂ device with even number of layers (N=2), the new sample G**. Not only could we observe a pronounced diode effect, but we could observe the by far largest rectification efficiency. On the one hand, we confirmed that, similarly to the Ising SC effect, the supercurrent rectification in NbSe₂ does not distinguish between even and odd number of layers, probably due to the weak interlayer electronic coupling. On the other hand, the reason for the varying rectification efficiency in the different devices remains unclear, see also the answer to the second-last question of Referee 1.

[Xi2016]: Xi, X., Wang, Z., Zhao, W. *et al.* Ising pairing in superconducting NbSe₂ atomic layers. *Nature Phys* **12**, 139–143 (2016). <https://doi.org/10.1038/nphys3538>.

2. *Since the diode behavior is driven by the out-of-plane magnetic field component, why is the experiment in Fig. 1 carried out in a magnetic field mostly directed in-plane? Would it be simpler and easier for understanding by directly show the effects and/or results under an out-of-plane magnetic field? In that case, the field values are meaningful.*

Our initial tests were conducted in cryostats where the magnetic field could be applied only in one direction. Coming from the recently discovered diode effect in Rashba systems, at that time we applied the field in-plane and perpendicular to the current (more precisely, we have mounted the sample holder at the cold finger stage in such a way to have the field oriented in that way with respect to the sample). However, in materials with valley-Zeeman-type spin-orbit the diode effect is mainly driven by B_z , as suggested by recent theoretical models (see e.g. the version 1 of the arxiv preprint Ref. [He2021], question 1 of Referee 1). This is exactly the reason why we performed dedicated experiments on sample F (Fig. 2), which are the main measurements of this work. These measurements were conducted in a dilution refrigerator system with a set of perpendicular coils and a rotator. It is on this system that we have the largest amount of data (see also the answer to Referee 1, question 3, about the reproducibility). These measurements made clear that both in-plane and out-of-plane fields play a role, but that the primary effect (a finite rectification Q) is driven by B_z only.

The Referee is right, the orientation of the field in the measurements shown in the (old) Fig.1 (sample B) was not the most convenient, and thus we performed the measurements on sample F (Fig.2). We took this comment very seriously: this is one of the reasons why we have measured our new sample (sample G) in the 1K cryostat setup, *rotating the sample holder in such a way to have the field perpendicular to the 2D crystal*. In this case, the field can be considered purely out-of-plane (a few-degree misalignment is in this case completely irrelevant, since effects of the in-plane fields require many hundreds of milliteslas, while in the new measurements on sample G we can have at most hundreds of microteslas in-plane). In this regime, we observe a clear diode effect. The results are in line with and further substantiate those reported in the original version of the manuscript.

Why the I_c curves in Fig. 1f have to be measured under two different temperatures? Would this induce the effect that the maximum I_c value is not at zero field?

First, we would like to point out that the new measurements we performed on sample G have eliminated this issue (measurements on sample B have been moved to the Supplementary Information).

Still, we shall answer the question of the Referee about Fig. 1f of the first version of the manuscript. The reason for the different temperatures is that we have measured first the regime of low to moderate field (from -2T to +2T) with high resolution - please consider that each data point corresponds to 2 IVs. After that, we started a new measurement session, with lower resolution but larger field range. Since, at this point, the liquid He level was much lower, we could not stabilize the temperature at 1.4 K anymore, the minimum temperature was then 1.6 K. The change in critical current due to this temperature difference is small, barely visible on the scale of Fig.1f. On the one hand, for reasons of transparency it is not reasonable to show measurements made at different temperatures together, even if the temperature difference is small. But, on the other hand, we found it useful to show the full range measurements together with the highly resolved measurement. Our solution was to normalize the two sets of data by the value $B=0T$, and to plot them together. This approach guaranteed transparency and

an overview of the full B dependence in one graph. In the new version of the manuscript, the previous Fig.1 c-g has been moved to the Supplementary Information. Instead, in the main text, measurements on sample G are presented at constant temperature.

Coming to the second question “*Would this induce the effect that the maximum I_c value is not at zero field?*”, the answer is no, because this effect is visible from the T=1.4K measurement series alone. Moreover, the new measurements on sample G still show the same effect at constant T.

What’s the T_c of the samples? Usually, it’s better to show RT curves (could be in the Supplementary Information) for a superconductor related experimental work.

The R(T) curves and T_c values were (and still are in the new version) shown and discussed in the Supplementary Information, see Section ” RESISTANCE AND CRITICAL CURRENT VERSUS TEMPERATURE”.

3. *The experiments of Fig. 2 measured on sample F were conducted by three-terminal configuration because one of the four electrodes stopped working after cool-down. Would it be possible to repeat the results of Fig. 2 in a non-broken sample? This is because three terminal configuration experiments would include same effects from electrode contacts, which may not be simply corrected by subtracting a normal resistance component (the contact resistance may not be linear with current or temperature).*

In the answer to the last question of Referee 1, we show some raw IV curves for sample F. It is clear that the source-drain contact resistance produces a linear slope. To a good approximation this resistance is independent of bias.

The important point here is that the only relevant information for our scope is the critical current, i.e., the bias at which the IV curve jumps to the higher resistance state. This critical current value is very easy to identify owing to the corresponding sharp discontinuity in the IV. It is also independent of the contact resistance and from the fact that we measure in 2-, 3-, or 4-terminal, provided that the discontinuity as a function of current bias is easily identifiable.

The method used to correct the field misalignment effect for the results in Fig. 2 is inconsistent with the main results in Fig. 1f. Figure 1f shows that the maximum I_c values are not at zero magnetic field. However, the authors corrected the misalignment effect for Fig. 2 by assuming that the maximum I_c values are at zero magnetic fields. This is really confusing.

The most direct answer we can provide is that the method is applied only to sample F (Fig. 2), where the critical current is maximal for B=0 (see Fig.2d,e), and not to sample B (Fig. 1f of the old version mentioned by the Referee).

However, we understand the concern of the Referee and we will try to answer more generally and comprehensively. Even if sample F showed the same phenomenology as sample B (namely, I_c maximal for a certain finite B_z , which we call $B_{max,Ic}$), our method would be still useful. The main problem we are trying to fix here is the fact that, due to misalignment, a finite B_{ip} produces a proportional additional B_z . As a result, a color plot as that shown in Fig.2a-c would have looked shear-distorted. What we need to

know here is the coefficient of proportionality between B_z and B_{ip} , which is ultimately the overall slope of the color plot in Fig.S3 of the old Supplement, now Fig.S7 in the new version.

This slope can be found easily and univocally, as described in the Supplement. If sample F showed a finite $B_{max,lc}$, then the color plot would have simply been shifted vertically by a certain small amount, whereas the shear distortion would have been always compensated correctly.

4. The vortex motion was mentioned as one of the possible mechanisms for SC diode effect by Toshiya Ideue and Yoshihiro Iwasa in Nature News and Views [Nature 584, 249 (2020)] for the work of Ref. 16. Later, a SC diode effect based on vortex motion mechanism was reported in Ref. 9 with the sample in an out-of-plane magnetic field, the same with the dominant field orientation in this work.

This is an important point to discuss. The concern of the Referee is legitimate: in the paper by Ando *et al.*, the sample is thick compared to the coherence length. Thus, an in-plane field is expected to produce vortices, the motion of which could lead to non-reciprocal Joule dissipation and thus a spurious (or extrinsic) diode effect. This, of course, is not what Ando *et al.* (nor what the subsequent theory models) meant to show, namely, a nonreciprocal behavior of the homogeneous superfluid. Unlike Ideue and Iwasa, we believe that one cannot conclude with certainty that the effect shown by Ando is extrinsic, namely, that results from vortices. We do agree with the Referee that it is highly desirable to exclude empirically any relevant role of vortices. For Rashba superconductors, this is comparatively easy to do: since the effect is triggered by the in-plane field, it is sufficient to work with samples much thinner than ξ to prevent nucleation of vortices in the first place. For valley-Zeeman superconductors, the situation is more complicated, since the field direction producing the diode effect is exactly the out-of-plane one, therefore one cannot avoid nucleation of vortices. For this reason, it is indeed useful to provide evidence that the role of vortices is unimportant.

In our case, one strong indication for this comes from the fact that for some samples the effect is so strong that the critical current increases (for one bias direction only) with the field. If the effect were driven by vortices, one would have expected a maximum of the critical current at zero field.

In general, one would expect emerging of vortices in a 2D superconductor in an out-of-plane magnetic field. Does the vortex motion play any or part of the role on the SC diode observed in this work? Although there are no symmetry breaking of vortex pinning sites (at least no artificial pinning sites) in the sample, however, the current flow with the two left current leads shown in Fig. 1b would not be uniformly straight along the sample channel (the channel is actually a square of 250 nm by 250 nm and is not long and 'narrow' considering the square geometry), as illustrated by the curved arrow line in Fig. 1b. This should lead to spatially symmetry breaking on the driving force to vortices. One possible experiment to exclude or confirm the possible vortex motion effect is to switch the current and the voltage leads. Applying current on the two right leads will reverse the symmetry breaking as compare to applying

current on the two left leads. This will reverse the diode polarity for the vortex motion induced diode effect.

In fact, in a vortex motion related effect, the Bean-Livingston surface barrier [C. P. Bean and J. D. Livingston, Phys. Rev. Lett. 12, 14 (1964); E. Zeldov. et al., Phys. Rev. Lett. 73, 1428 (1994)] will lead to vortices entering and existing the sample not exactly at zero magnetic field. This effect is usually stronger for narrower samples. Would this effect related to the result of the maximum value of the I_c curves in Fig. 1f is not at zero magnetic field? Such effect can be easily excluded or confirmed by comparing results between two field sweeps (increasing and decreasing). For example, for a given current polarity, the maximum I_c value would be at positive (or negative) fields for increasing (or decreasing) fields, respectively. To be noted, reversing the field sweep direction would NOT reverse the diode polarity which is determined by the spatially symmetry-breaking.

Before answering the main question of the Referee (“*Does the vortex motion play any or part of the role on the SC diode observed in this work?*”), we would like to discuss why we believe that the position of the electrodes within the two large SC banks (connected by the constriction) is irrelevant. In fact, each of the two banks is a superconductor where the supercurrent flows with current density much lower than the critical value (which is reached first in the narrow constriction). Therefore, the electrochemical potential will be exactly the same in every point of the bank, irrespective of the location of the electrode. The key here is that the constriction is much narrower than the rest of the NbSe₂ crystal. The yellow line in Fig.1b is only meant to illustrate the layout of source and drain, and not the current density distribution within the constriction. Also, the width of the constriction is much smaller than the Pearl length λ^2/d = tens of μm (the constriction width is comparable to λ) in NbSe₂, therefore the supercurrent density distribution is necessarily homogeneous.

Now we turn to the answer to the main question about the role of vortices. We believe that vortices do not play any significant role in determining the diode effect owing to the following reasons:

(i) The presence of vortices generate dissipation, when subjected to finite current. As a consequence, IV-characteristics usually display an onset of flux creep below I_c . In principle, if the rate of vortices entering the constriction is different for the two directions of the current bias, the Joule dissipation would also be polarity-dependent, and so it would be the electron temperature and, finally, the critical current I_c .

As a matter of fact, however, we do not observe a measurable onset of dissipation in the IVs until we reach the critical current, and this up to at least the field for maximal Q , i.e. $B_{\text{max},Q}$. As an example, in Fig.R3 we show the IV characteristics for sample D (where 4-terminal IV-characteristics are available). In this case, as described in the manuscript, the field is applied mainly in-plane, with an important out-of-plane component (of the order of several tens of mT).

Notice that there is no measurable flux creep within a voltage scale of 500 nV (noise level) and at the critical current the voltage emerges abruptly from the noise floor.

Fig.R3: (a) 4-terminal IV characteristics for sample D, measured at 1.3 K at an applied field of $B=1.25$ T, approximately the value providing the maximum rectification Q for this sample, see Supplementary Figure S4 of the new version of the manuscript. (b) Zoom-in (y-scale magnified 250 times). On a sub- μV scale, there is no measurable dissipation due to vortices, the dissipation emerges abruptly from the noise floor. These graphs have been added to the new Supplementary Information, Fig. S6.

(ii) If the diode effect were due to the asymmetry in the barrier for entering the sample, it would be extremely T -dependent (as is λ and thus the vortex barrier) while approaching T_c . Instead, the T -dependence can be even non-monotonic (sample D and F, see Fig.3 of the main text) and in general not so pronounced even relatively close to T_c (see Fig. 3 of the main text). On the other hand, non-monotonicity is expected from theory (see e.g. [Ilic2021], answer to Question 1 of Referee 1).

(iii) Field sweep direction. As pointed out by the Referee, if the diode effect were due to the asymmetry of the edge barriers for entering/leaving the constriction, then this would produce a relevant hysteresis in the field sweep, and vortices would be trapped in the constriction at zero field. To test this hypothesis, in the last measurement session on sample G we swept B_z back and forth multiple times, as suggested by the Referee. As is visible in Fig. R4 below, the sweep direction does not play any role: the $I_c^+(B_z)$, $I_c^-(B_z)$ and $Q(B_z)$ curves fall on top of each other within the experimental uncertainty.

We also note that, if vortex barriers were relevant, then the effect of the out of plane field would be not only hysteretic, but also nonlinear: as long as the first vortex enters the constriction, it would strongly affect the barrier for the subsequent vortices. Instead, the $Q(B_z)$ curves appears smooth and nearly linear at small fields, see averaged $Q(B_z)$ curve in Fig.1g of the main text.

Fig.R4: (a) Four measurements of Q (sample G, $T=1.3$ K) as a function of B_z , where B_z is swept in different directions, as indicated in the legend. The measurement sequence is indicated in the legend (top first). (b) Similar sequence of measurements, but with inverted sign for B_z . This sequence was measured after that in (a) and after a demagnetization routine for the cryostat magnet. These are raw data (no instrumental offset subtraction), not averaged (therefore the data point scatter is larger than in the new Fig.1g). It is clear that the sweep direction plays no evident role. The two graphs are merged into the new Fig.S1b.

(iv) If the diode effect were only due to vortices, the in-plane \mathbf{B}_{ip} field would not make the effect of B_z asymmetric, as we observe in the experiment. *A fortiori*, the \mathbf{B}_{ip} sign and orientation would be totally irrelevant, in contrast to results shown in Fig.2 of the main text.

The above comment does NOT mean the author's conclusion of valley-Zeeman SOI induced diode effect in NbSe2 nanowires could be wrong. I recommend the authors to discuss and/or exam the possible role from vortex motion because the diode effect may not originate from a single mechanism.

The concern of the Referee is legitimate: unlike Rashba systems, in SC with valley Zeeman SOI the diode effect is driven by the out-of-plane field, which unavoidably introduces vortices. It is thus important to understand their role, although this check is not always provided in the experimental literature (leading to the comment by Toshiya Ideue and Yoshihiro Iwasa in Nature News&Views [Nature 584, 249 (2020)] mentioned by the Referee).

We believe that the arguments (i-iv) discussed above rule out any significant role of vortices for the emergence of the diode effect *in our experiments*. This does not mean that they do not play any role in general. We argue, for example, that the erratic process of vortices which are entering/getting pinned into/exiting the constriction might be (at least partially) responsible for the scatter of the data points for I_c (and thus Q) highlighted for example by Referee 1.

In summary, the manuscript presents a very interesting and novel SC diode effect in few-layer NbSe₂ nanowires driven by out-of-plane magnetic fields. Given that the SC diode effect is significant for both future application and new physics, I will recommend publication of this work in Nature Communications if the authors could address my above comments/confusing properly.

We thank the Referee for her/his positive comments. We believe that our additional efforts, motivated by the constructive criticism of both Referees, have significantly improved the manuscript and dissipated the legitimate doubts.

REVIEWERS' COMMENTS

Reviewer #1 (Remarks to the Author):

In my view the authors have responded satisfactorily to the questions raised by all two reviewers. While not all details of the observations are explained yet, the main message is important, clearly expressed and supported by the data. Hence, I support publication in Nature Communications.

Reviewer #2 (Remarks to the Author):

The authors successfully addressed most of my comments by new experiments and thorough discussions.

However, I am not agree on their statements in the revised manuscript (second paragraph in page3) regarding ruling out the possible role of vortices based on the result of the maximum I_c at nonzero field. Their statement "the maximum critical current would then be always observed in absence of vortices, i.e., at $B_z = 0$ " is not always true. For example, at zero field the electrical current can lead to self-induced vortex and antivortex pairs moving oppositely in the sample. In certain cases the I_c at a nonzero field can be higher than that at the zero field due to vortex motion, e.g. in PRL 111, 067001 (2013). Again, I'm not arguing the SC diode realized in this work is due to vortex motion, however, completely ruling out the role of vortices in a diode effect from an out-of-plane field could be challenging. Although the authors' answers regarding the possible role of vortices are still discussible, considering the fast development of SC diodes I do not want to rise more questions on this because these discussions could cause a lot of delay on the publication of this work. The diode effect demonstrated in the few layer NbSe₂ is clear, which is an original contribution to the research field. For a fast developing research field, it would be OK to leave some open questions, e.g., the mechanism of the field-free Josephson diode in the recently published Nature 604, 653 (2022). These open questions would attract more interests and future investigations.

One more technical recommendation: For the IV curves shown in this work, it seems they were not measured by simply/directly using a current source or a voltage source (there are two voltage/resistance values for a given current value near I_c), so please state in the Method section on the detailed experimental method, e.g., any electrical circuits used in the transport measurement loop.

Reply to the Referees.

Reviewer #1 (Remarks to the Author):

In my view the authors have responded satisfactorily to the questions raised by all two reviewers. While not all details of the observations are explained yet, the main message is important, clearly expressed and supported by the data. Hence, I support publication in Nature Communications.

We thank the Referee for her/his comments.

Reviewer #2 (Remarks to the Author):

The authors successfully addressed most of my comments by new experiments and thorough discussions.

We would like to thank the Referee for her/his valuable comments.

However, I am not agree on their statements in the revised manuscript (second paragraph in page3) regarding ruling out the possible role of vortices based on the result of the maximum I_c at nonzero field. Their statement “the maximum critical current would then be always observed in absence of vortices, i.e., at $B_z = 0$ ” is not always true. For example, at zero field the electrical current can lead to self-induced vortex and antivortex pairs moving oppositely in the sample. In certain cases the I_c at a nonzero field can be higher than that at the zero field due to vortex motion, e.g. in PRL 111, 067001 (2013). Again, I’m not arguing the SC diode realized in this work is due to vortex motion, however, completely ruling out the role of vortices in a diode effect from an out-of-plane field could be challenging. Although the authors’ answers regarding the possible role of vortices are still discussible, considering the fast development of SC diodes I do not want to rise more questions on this because these discussions could cause a lot of delay on the publication of this work. The diode effect demonstrated in the few layer NbSe2 is clear, which is an original contribution to the research field. For a fast developing research field, it would be OK to leave some open questions, e.g., the mechanism of the field-free Josephson diode in the recently published Nature 604, 653 (2022). These open questions would attract more interests and future investigations.

We discussed further the possible role of vortices, also in light of a very recent preprint work (Hou et al.). This new discussion mainly affects one paragraph at page 3 of the new manuscript, see pdf with changes marked in red.

One more technical recommendation: For the IV curves shown in this work, it seems they were not measured by simply/directly using a current source or a voltage source (there are two voltage/resistance values for a given current value near I_c), so please state in the Method section on the detailed experimental method, e.g., any electrical circuits used in the transport measurement loop.

We modified the Methods accordingly.